

# Representativeness of Total Column Water Vapour Retrievals from Instruments on Polar Orbiting Satellites

Hannes Diedrich[1], Falco Wittchen[1], René Preusker[1], and Jürgen Fischer[1]

[1]Institut für Weltraumwissenchaften, Freie Universität Berlin, Carl-Heinrich-Becker-Weg 6-10, 12165 Berlin

*Correspondence to:* Hannes Diedrich (hannes.diedrich@wew.fu-berlin.de)

**Abstract.** The remote sensing of Total Column Water Vapour (TCWV) from polar orbiting, sun-synchronous satellite spectrometers such as the Medium Resolution Imaging Spectrometer (MERIS) on board of ENVISAT and the Moderate Imaging Spectroradiometer (MODIS) on board of Aqua and Terra enables observations on a high spatial resolution and a high accuracy over land surfaces. The observations serve studies about small scale variations of water vapour as well as the detection of local

and global trends. However, depending on the swath width of the sensor, the temporal sampling is low and the observations of TCWV are limited to cloud-free land scenes.

This study quantifies the representativeness of a single TCWV observation at the time of the satellite overpass under cloud-free conditions by investigating the diurnal cycle of TCWV using 9 years of a 2-hourly TCWV data set from global GNSS (Global Navigation Satellite Systems) stations. It turns out, that the TCWV observed at 10:30 local time (LT) is generally

lower than the daily mean TCWV by 0.65 mm (4 %) on average for cloud-free cases. Averaging over all GNSS stations, the monthly mean TCWV at 10:30 LT, constrained to cases that are cloud-free, is by 5 mm (25 %) lower than the monthly mean TCWV at 10:30 LT of all cases. Additionally, the diurnal variability of TCWV is assessed. For the majority of GNSS stations, the amplitude of the averaged diurnal cycle ranges between 1 % and 5 % of the daily mean with a minimum between 6 LT and 10 LT and maximum between 16 LT and 20 LT. However, a high variability of TCWV on an individual day is detected. On

average, the TCWV varies by 15 % around the daily mean.

## 1 Introduction

Water vapour plays a key role in the hydrological cycle of the earth's atmosphere. The Total Column Water Vapour (TCWV) is a good indicator/tracer of atmospheric transport of water vapour. The diurnal cycle of TCWV over land is influenced by evapotranspiration as a source, condensation and precipitation as sinks, and additionally by atmospheric advection (Trenberth,

1999). Over the years, multiple techniques have been established to determine the TCWV from ground and from space. TCWV from measurements of radiosondes, microwave radiometers and Global Navigation Satellite Systems (GNSS) receivers are examples for sophisticated ground based sources of TCWV values on a high temporal resolution that are hardly affected by clouds (e.g. RS: Seidel et al. (2009), MWR: Turner et al. (2007), GNSS: Dick et al. (2001)). Unfortunately, the ground based measurements do not resolve the spatial structures of water vapour fields. Further, they are usually limited to land areas.

However, satellite remote sensing allows observations of TCWV on a high spatial resolution. Over land surfaces, TCWV





derived from radiance measurements in the near-infrared (NIR) from space borne spectrometers meets the requirement needed for weather forecasts and climate studies, due to high accuracy and high spatial resolution (up to 300 m) of the TCWV products. Observations from the MEdium Resolution Imaging Spectrometer (MERIS) (Bennartz and Fischer, 2001; Lindstrot et al., 2012) on ENVISAT and the Moderate Resolution Imaging Spectroradiometer (MODIS) on Aqua and Terra (Gao and Kaufman, 2003;

Diedrich et al., 2015) can provide long time series of TCWV. These data sets such as described by Lindstrot et al. (2014) benefit global trend analysis or investigations of small scale phenomena as described by Carbajal Henken et al. (2015). However, there are two major drawbacks of observations by polar orbiting satellites:

- Most areas are sampled only once per day or even less depending on the latitude and the swath width of the instrument.

- Clouds are opaque in the visible and NIR spectrum. Consequently, the observations of TCWV are limited to cloud-free
areas.

For the observation of the diurnal variability of TCWV ground based microwave radiometer and GNSS measurements are appropriate. The influence of clouds and precipitation can be neglected. TCWV can be derived from measurements of the Zenith Path Delay (ZPD) of ground based GNSS receivers even under cloudy conditions and on temporal resolutions up to a few minutes. 9 years of 2-hourly TCWV data derived from GNSS measurements has been used in order to answer the following

questions:

1. How large is the variability of the TCWV in comparison to the daily mean TCWV?

2. How representative is the TCWV at the time of the satellite overpass to the daily mean TCWV?

3. How representative is the climatology of the cloud-free TCWV to the TCWV climatology including cloudy conditions at the time of the satellite overpass?

There are few studies that examine the diurnal cycle of water vapour such as Li et al. (2007), Ortiz de Galisteo et al. (2011), Radhakrishna et al. (2015). However, these works usually focus on certain regions. In this study we like to highlight the potential of a global TCWV data set and present a global analysis of the diurnal cycle of TCWV. We want to give an overview of the variability of TCWV that is needed for the interpretation of water vapour fields derived from remote sensing.

## 2   Satellite TCWV datasets

As mentioned above, the daily coverage of imaging spectrometers on sun-synchronous polar orbiting satellites is limited by the field of view of the specific instrument. MERIS on ENVISAT has a swath width of 1150 km which leads to global coverage in about 2-3 days. MODIS on Terra scans the earth in 1-2 days with a swath width of 2330 km. Consequently, in the lower latitudes observations take place only once every 3 days, in the middle latitudes about once per day. Information about the daily cycle of TCWV can not be retrieved from this kind of observations. However, climate monitoring requires trend analysis which

is performed with the aid of TCWV from space borne spectrometers due to the global coverage. ENVISAT and Terra cross the





equator at about 10:30 a.m. local time, both at descending note. TCWV retrievals that are based on radiance measurements in the NIR (Diedrich et al., 2015; Lindstrot et al., 2012) are limited to cloud-free areas where high accuracies can only provided over land surfaces (Diedrich et al., 2013).

## 3   GNSS TCWV dataset

Basis of our investigation is a 2-hourly TCWV data set from Wang et al. (2007) for the years 2003 to 2011. The TCWV was derived from ground based GNSS measurements of Zenith Path Delay (ZPD) using three different resources, including the International GNSS (Global Navigation Satellite Systems) Service (IGS) tropospheric products, U.S. SuomiNet (UCAR/COSMIC) products and Japanese GEONET (GNSS Earth Observation Network) data (Earth Observing Laboratory, 2011). All GNSS TCWV retrievals are based on the same procedure, explained shortly in the following. The TCWV is derived from the delay

of the GNSS signal, that is introduced by interactions with the atmosphere. By subtracting the ionospheric and hydrostatic attenuation, and accounting for the elevation angle of the satellites, the Zenith Wet Delay (ZWD) can be approximated, that is in the range of a few centimetres (Bevis et al., 1992). Subsequently, the ZWD is converted to TCWV. Although there are a number of error influences, the uncertainty of TCWV derived from GNSS is about 1-2 mm (Ning et al., 2015).

TCWV observations from about 1000 global distributed stations for the period between 1995 to 2011 are available in the

data set. However, the majority of stations do not contribute continuously over the whole time and the number of stations increases with time. In order to have a relatively complete time series of TCWV for each station, we selected only locations that provided at least 5 years of data in the period of 2003 and 2011. Figure 1 shows the spatial distribution and the elevation of the selected 296 stations. There is a high density in Europe and the USA and only a few stations in South America, Africa and Central Asia. Nevertheless, the spectrum of the locations is diverse. The stations are located in very dry and humid conditions

at continental and coastal locations range from sea level to 3600 m. Besides other influences, the diurnal cycle of water vapour in the lower troposphere is hypothetically linked to the diurnal cycle of temperature that is in turn mainly driven by the position of the sun. Therefore, we converted the time information in the GNSS data set (given in UTC) to the local time (LT) that is used hereafter and is derived as follows:

$$LT = UTC + (\varphi/15) \; , \;\; \varphi \, \epsilon \{-180, ..., 180\} \, ,\tag{1}$$

where $\varphi$ is the longitude of the location of the corresponding station.

## 4   Diurnal Cycle of TCWV

The main part of the column integrated water vapour is located in the boundary layer. Consequently, TCWV can represent the processes related to water vapour that take place in the lower troposphere. There are several mechanisms that influence the TCWV. The most important ones are: Evaporation and condensation, large-scale and local advection of moist or dry air. Con-

sidering averages over a large number of days, the large scale advective part is usually represented at all times of a day, leaving only the variations that are connected to the diurnal cycle in air temperature. With increasing surface temperature over the



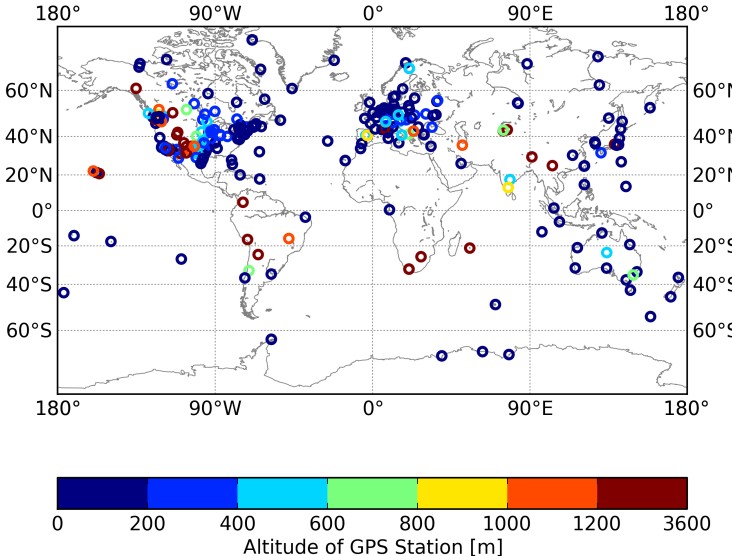

**Figure 1.** Global distribution and height in meters of selected GNSS stations. Note that the last height class contains all stations between 1200 m and 3600 m elevation.

day, the evaporation can increase. At night water vapour condensates and consequently, TCWV decreases on average. Another influence is the advection of humid or dry air masses. Winds originate from synoptic situations or orographic circumstances. The wind is generally higher at daytime because of convection that in turn will also influence the water vapour amount. Local geographic conditions can result in circulation patterns that occur almost every day such as land and sea breeze and mountain

breeze. The differential warming between land and ocean carries moisture onshore at daytime. At night-time this circulation is reversed due to the faster cooling of the land surfaces. The climatic and geographic conditions are various for the selected stations. In some cases, these influences are dependent on the time of the year due to the annual variability of the circulation pattern. To analyse every single station concerning its diurnal cycle would exceed the frame of this paper. Consequently, a statistical approach of the evaluation of the diurnal cycle is presented in the following.

An evaluation of the diurnal variability of the TCWV anomaly from the daily mean for the 296 considered stations averaged over 9 years (2003-2011) is represented in Figure 2 as box plot. Black boxes and whiskers indicate a histogram of the TCWV anomaly including all stations for each 2-hour time step. The inner quartile range (IQR) is varying between +2 % and -2 % of the daily mean TCWV (indicated by the horizontal dashed line). There is a significant minimum of the station mean TCWV (indicated by the horizontal bar in the boxes) between 6 LT and 10 LT and a maximum between 16 LT and 20 LT with an

amplitude about 1 %. The variation of 95 % of the stations is ranging between +5 % and -5 %.

In order to analyse the influence of the location to the averaged daily cycle of TCWV, two subgroups of stations are selected: 52 Coastal stations that are situated within 5 km of a coast of the ocean or large lakes and below 800 m elevation (indicated orange in Figure 2 and 3), and 44 high GNSS stations that are situated above 800 m (indicated green). The coloured lines in





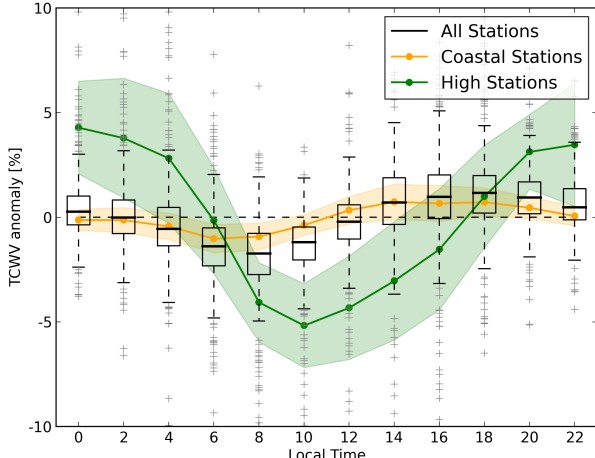

**Figure 2.** Boxplot of TCWV anomalies from the daily mean in % for all stations and the period between 2003 and 2011. The range of the black boxes indicate the interquartile range (IQR), containing 50 % of the data points (each point represents one stations). The horizontal bar within the boxes represent the median; vertical bars (whiskers) indicate the reach of 95 % of the data points; and grey pluses show outliers. Green line: Mean daily cycle of TCWV of all high stations (greater than 800 m altitude); Orange line: Mean daily cycle of TCWV for coastal stations. See text for detailed description.

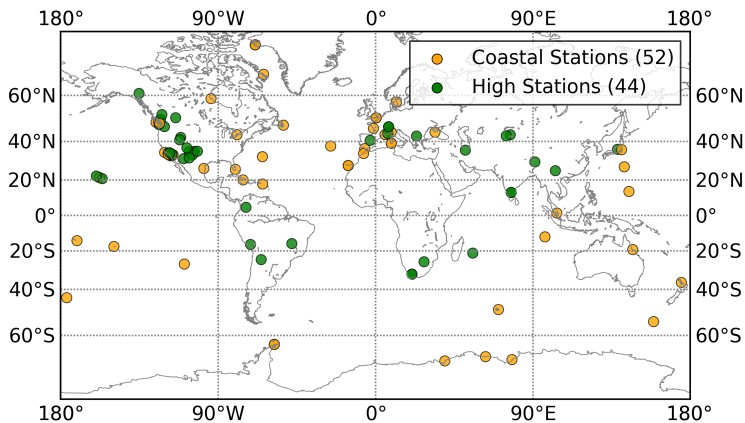

**Figure 3.** Location of the stations that where selected for coastal (orange) and high stations (green).

Figure 2 show the mean diurnal TCWV anomaly of the two groups including the range of the standard deviation as coloured shading. The diurnal variability of the high stations is most pronounced in this comparison, peaking at 0 LT and 10 LT with an amplitude of about 5 %. The reasons for this particular shape could be manifold depending on e.g. the local circumstances of the terrain and availability of humidity. A possible explanation for the large amplitude of the diurnal cycle of TCWV anomalies could be the larger daily variation of air temperature in mountain areas resulting from enhanced warming of the slopes at




daytime and enhanced cooling at night due to the higher surface area. In order to find out the true reasons of the different diurnal cycles of high stations, other data has to be taken into account, such as air temperature.

The diurnal mean TCWV anomaly of coastal stations is following the overall mean TCWV anomaly of about 1 %. Here, the sea breeze could be an explanation for the peak times. The flow of humid air from the ocean is maximal in the afternoon and minimal in the morning. The relative amplitude of the diurnal cycle of TCWV is low in comparison to other stations presumably because of the general high humidity at the coast.

The variation of the averaged diurnal cycle of TCWV anomalies between the high stations is larger than between the coastal stations, represented by the standard deviation. This results potentially from the large variation of locations of the high stations.

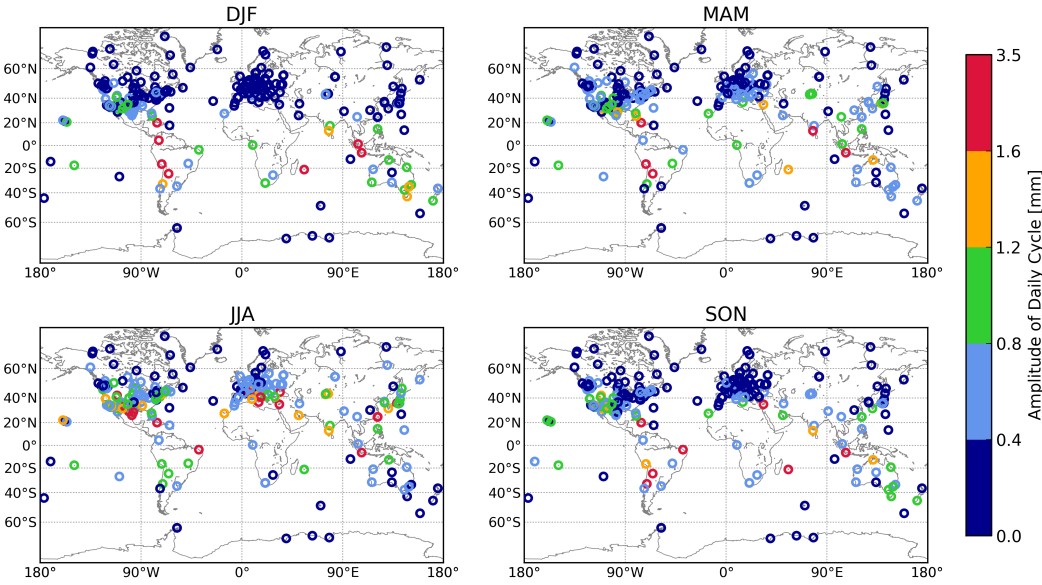

**Figure 4.** Season averaged amplitude of the daily cycle of TCWV for each GNSS station in mm.

In order to get the global view on the averaged daily variability of TCWV we derived measures that indicate the shape and amplitude of the mean diurnal cycle of each GNSS station. In Figure 4 the season-averaged amplitude, meaning the difference between the maximal TCWV and the daily mean TCWV, of the daily cycle of TCWV is plotted for each station. The first obvious feature is that the amplitude of the diurnal cycle is increasing with decreasing latitude and the maximum of the zonal mean amplitude is moving north in northern summer and south in northern winter. This can be explained by the annual variation of the lower tropospheric temperature. Furthermore, the amplitude ranges from 0.1 mm to 0.8 mm in the middle and high latitude to 3.5 mm for stations in the tropics and mid-west of the USA. Considering the potential range of TCWV in the atmosphere between 1 mm and 70 mm this range seams rather low. In order to study how much the daily cycle constitutes to the daily mean TCWV, the anomalies from the daily mean are presented in Figure 5, averaged again over the seasons. It shows, that for most of the stations the daily variability is only in the range of 1-5 %. For most of the stations with higher altitudes the




anomalies are increased and range up to 31 % that is consistent with the findings in Figure 2. In general, the amplitude of the daily anomalies is not following the global temperature distribution.

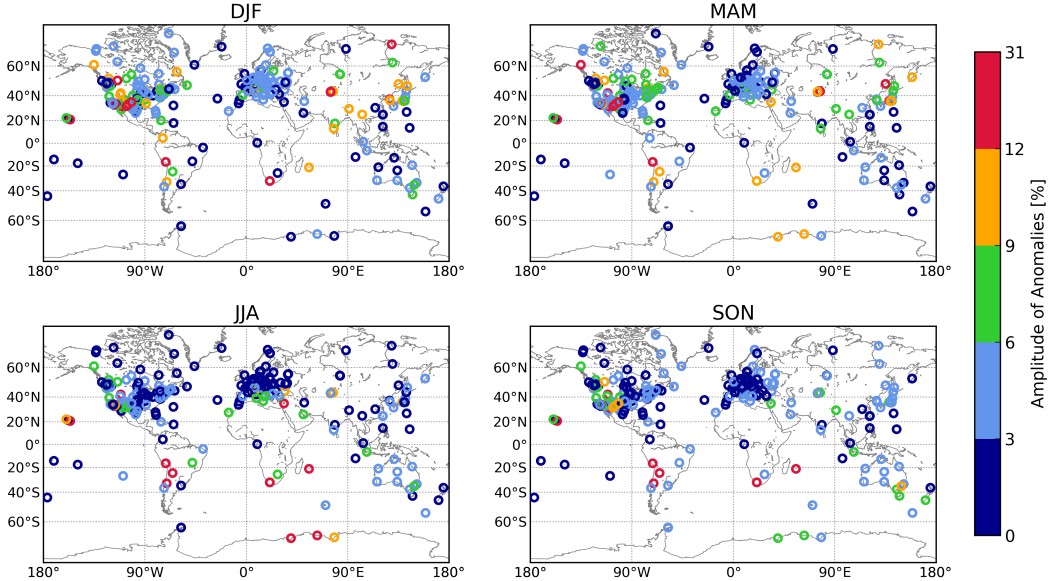

**Figure 5.** Season averaged amplitude relative to the daily cycle of TCWV for each GNSS station in %.

Another interesting feature of the diurnal cycle of TCWV is the time, when the TCWV reaches its maximum or minimum. In Figure 6 and 7 the local time of occurrence of the maximum TCWV respectively the minimum TCWV of the averaged
daily cycle for every season is shown. At the majority of the stations a similar shape of the diurnal cycle appears resulting in the same times of the maximum and the minimum in each season. This is demonstrated by the dominating reddish and blue colors on the maximum plot and dominating green and orange colors in the minimum plot. It leads to a mean daily TCWV variation that is minimal in the morning between 4 LT and 12 LT and that is maximal at night between 16 LT and 4 LT that again is consistent with the mean daily cycle for all stations in Figure 2. However, there are stations with different characters,
e.g. in the Rocky Mountains that peak between 4 LT and 8 LT for the maximum and between 16 and 20 LT for the minimum. This shape is reverse to the other stations. At the ENVISAT and Terra overpass the average TCWV for the majority of stations is still slightly below the daily mean. This is the main reason for the negative bias between the TCWV derived at 10:30 LT and the daily mean (next section).

In general, the variation of the averaged diurnal cycle of TCWV between the stations is large concerning the time of the
maximum or minimum and the amplitude and is mainly dependent on the location of the GNSS station. This is consistent with Figure 2, where the spread of the standard deviations, boxes and whiskers indicate that the averaged diurnal cycle is varying by more than ±10 % between the stations. The averaged amplitude is only in the range of a few percent of the mean TCWV.

However, this information does not quantify the individual daily variability of the TCWV. Figure 8 shows the TCWV for 16 days in January 2003 for the GNSS station Potsdam as dashed line. Additively, the daily mean TCWV is plotted. For better



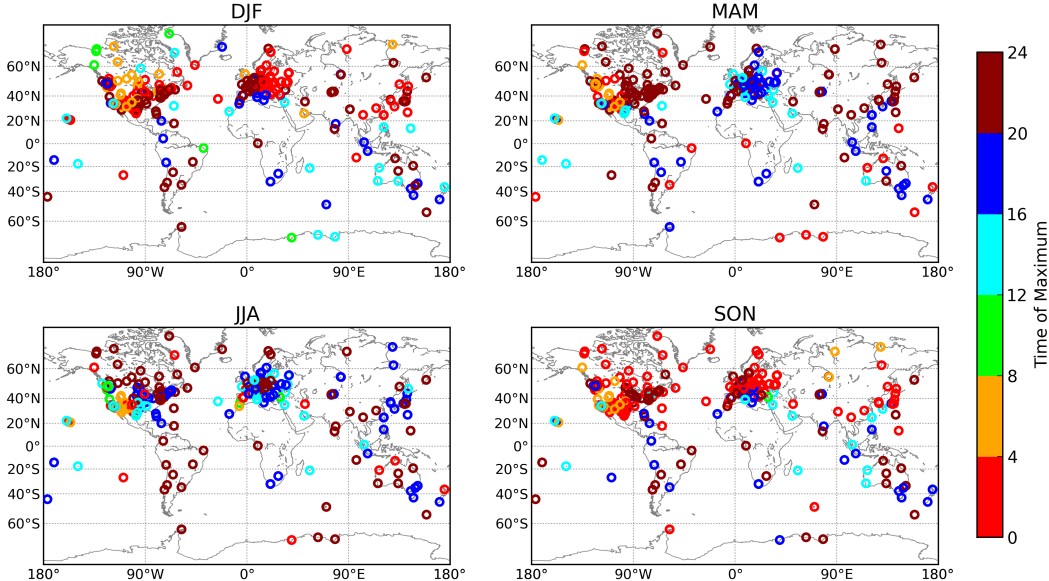

**Figure 6.** Local time of occurrence of the maximum of the seasonal averaged daily cycle.

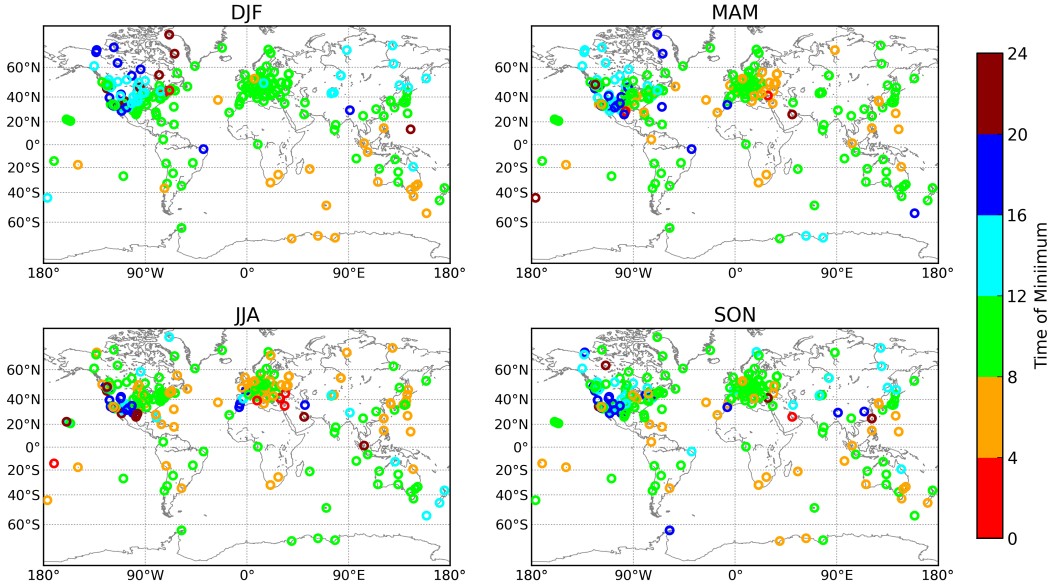

**Figure 7.** Local time of occurrence of the minimum of the seasonal averaged daily cycle.

visibility, the positive difference between the curves are filled red and the negative areas are filled blue. The comparison reveals that the daily cycle of TCWV is different for every day. In order to quantify this variability, the daily standard deviation (DSD) of the TCWV as anomaly from the daily mean, averaged for each station is plotted on the right panel in Figure 9 as frequency



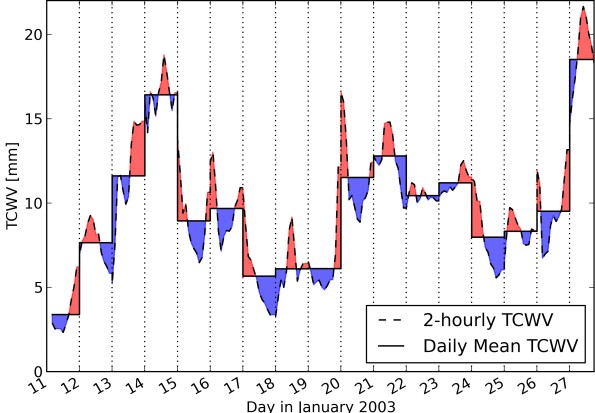

**Figure 8.** GNSS TCWV vor 16 days in January 2003 at the station Potsdam (POTS) in blue. Daily average TCWV in black. Anomalies are coloured (positive: red, negative:red).

density plot. The DSD for each station ranges between 5 % and 35 % . The median is at 15.1 % (equivalent to 2 mm TCWV; indicated by the dashed vertical line) and 80 % of the stations show a DSD between 11 % and 21 % (indicated by the green color). Additionally, the location of the percentile groups is plotted in the left panel of Figure 9. GNSS stations with low DSD (10th percentile; indicated blue) are generally distributed in the tropical region. This region is known to have a low daily

5  variability of temperature and humidity. The outliers with high SDS (red) are not limited to special climatological regions.

Summarizing, the averaged anomaly of the daily mean TCWV varies only between 1 % and 5 % for the majority of stations. This is an important information for the interpretation of climatologies of TCWV from observations of polar orbiting satellites (see next chapter). However, the variability around the daily mean for an individual day is significantly higher, on average up to 35 % .

## 10  5  Representativeness of TCWV at 10:30 LT to daily mean TCWV

Whether one observation at the overpass time of the satellite is representative for the daily mean TCWV was investigated. The bias between the TCWV (observed at 10:30 LT) and the daily mean TCWV (for all days that were cloud-free at 10:30 LT) was averaged for each GNSS station and shown in Figure 10. The right plot represents the distribution of frequency of the bias for each station. Blue bars indicate the bias classes that are within the 10th percentile of the distribution and red bars above

15  the 90th percentile. The spatial distribution of the GNSS stations is shown in the left plot. A negative bias indicates that the 10:30 LT TCWV is lower than the daily mean. Here, the number of considered stations is reduced to 202 because some stations where mostly cloud covered at 10:30 LT. Figure 10 shows a station-mean bias of -0.63 mm (illustrated by the vertical dashed line in the histogram) that is within the measurement accuracy of the GNSS measurements and within the uncertainty ranges of a typical TCWV retrieval from observations in the NIR (Diedrich et al., 2013). The station with low biases are distributed in

20  the tropical region, where the variability of TCWV is smaller than in other regions. In general, negative biases appear at nearly




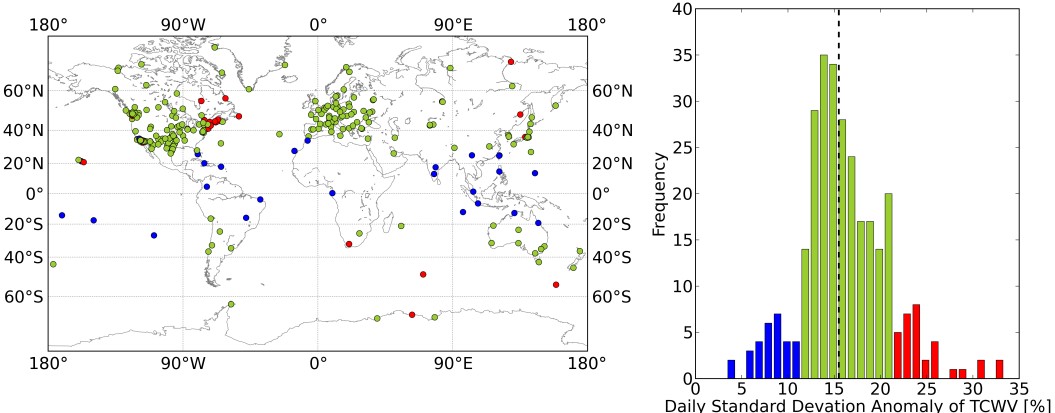

**Figure 9.** Mean daily standard deviation anomaly from the daily mean TCWV in %. Right plot: Histogram of the bias, blue bars indicate the classes that are in the 10th percentile, red bars indicate classes that are above the 90th percentile; the vertical dashed line illustrates the position of the station-median standard deviation anomaly. Left plot: Spatial distribution of stations coloured in the three percentile classes.

all stations. This is consistent with the findings of the last chapter, where the averaged anomaly of the diurnal cycle of TCWV is negative at 10:30 LT.

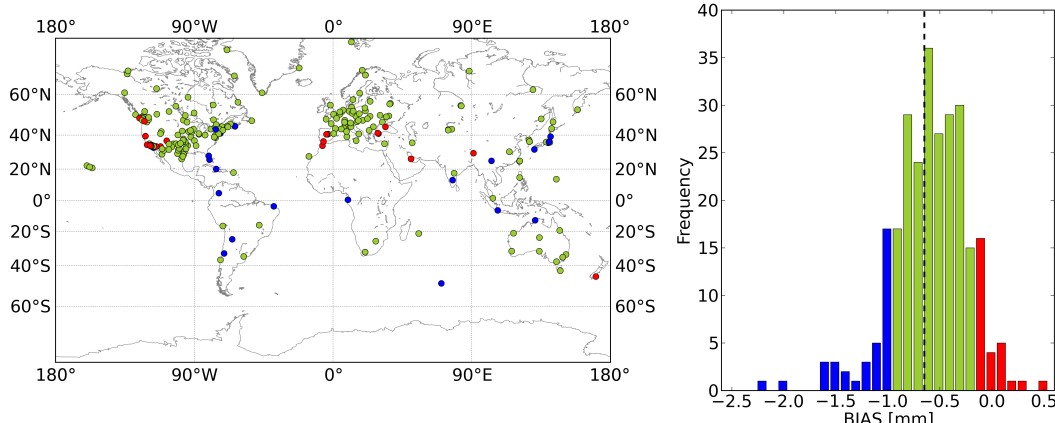

**Figure 10.** Bias for each station between the TCWV observed at 10:30 LT and the daily mean TCVW for all cases, that were cloud-free at 10:30 local time. Right plot: Histogram of the bias, blue bars indicate the classes that are in the 10th percentile, red bars indicate classes that are above the 90th percentile; the vertical dashed line illustrates the position of the station-median bias. Left plot: Spatial distribution of stations coloured with the three percentile classes.



## 6 Representativeness of cloud-free monthly-mean TCWV to monthly-mean TCWV at 10:30 LT

The fact that TCWV derived from instruments like MERIS and MODIS is limited to cloud-free areas has to be accounted for in the interpretation of trend analysis. Figure 11 shows monthly mean TCWV derived at 10:30 LT of all considered GNSS stations for the period 2003-2011 in blue including all cases and cloud-free cases in red. There is a clear difference in TCWV between all

cases and non-cloudy cases. On average, TCWV is about 25 % (5 mm) higher for all scenes than for clear scenes. This increase in TCWV for cloudy cases has been detected also in the study by Gaffen and Elliott (1993). Concerning climatological studies of absolute TCWV values, the TCWV observed at cloud-free cases is not representative for the TCWV including all cases. Nevertheless, the cloud-free and all-case TCWV are highly correlated.

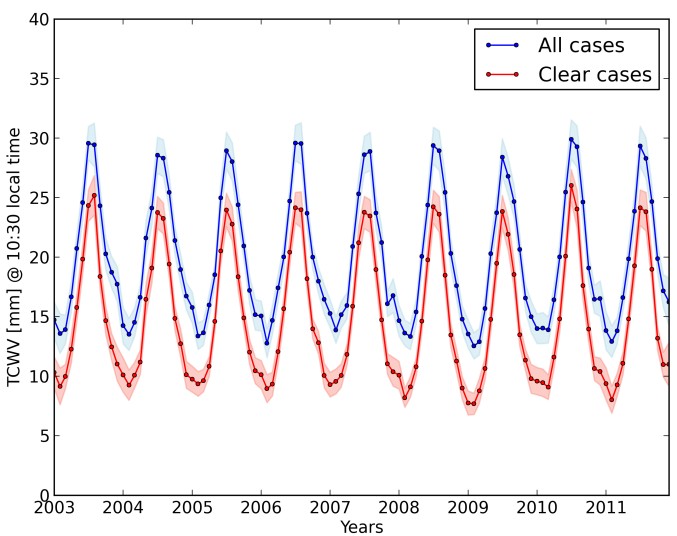

**Figure 11.** Monthly mean TCWV of all selected stations for cloudy scenes in blue and non-cloudy scenes in red lines.The shading indicates the 95 % -significance interval.

## 7 Conclusions

In this investigation the representativeness of the TCWV derived from imaging spectrometers that measure radiance in the NIR on polar orbiting satellites is discussed. A TCWV data set derived from GNSS delay measurements that is hardly influenced by clouds has been used. It turns out that on average the TCWV observed at 10:30 LT on cloud-free cases is generally lower than the daily mean TCWV. The bias of -0.65 mm (-4 % ) is in the range of the mean amplitude of the diurnal cycle. The monthly mean TCWV observed at 10:30 LT constrained to cloud-free cases is significant lower than the monthly mean TCWV of all

cases by about 25 % (5 mm). Nevertheless, the diurnal cycle is only a few percent of the daily mean TCWV for most of the stations. On average, for the majority of stations TCWV peaks at the evening and is minimal in the early morning local time. However, the variability on an individual day is much higher. The daily standard deviation averaged for every station is about



15 % of the daily mean. Summarizing, the biggest influence on the representativeness of observed TCWV from polar orbiting satellites is the constraint to cloud-free areas. The time of observation is a minor factor and in case of the MODIS and MERIS overpass negligible when averaging over time. The used GNSS-TCWV data set, collocated with cloud information data from satellites, offers a lot of potential for studies concerning for example the interaction between clouds and water vapour. As

5   precipitation is also strongly influencing the water transport and consequently TCWV, the collocation of rain gouge data can also serve a more detailed view of the reasons for the individual diurnal viability of TCWV.

*Acknowledgements.* This study was performed in the framework of the BMWi project WaDaMo and supported by ESA (European Space Agency) project SEOM (Scientific Exploitation of Operational Missions). The authors would like to thank Galina Dick from GFZ (Geoforschungszentrum) Potsdam for the input concerning GNSS TCWV retrievals and interpretation.





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
