# Peer review of "Representativeness of Total Column Water Vapour Retrievals from Instruments on Polar Orbiting Satellites"

_Atmospheric Chemistry and Physics, 2016_

## Referee Comment (RC1) · Anonymous Referee #1 · 6 May 2016

The manuscript falls into the scope of ACP and has the scientific quality to be published. It is very well written and organized and although there are in the literature many comparison of PWV data from satellites and GPS, none of them from the point of view of the authors, who make a great scientific contribution to validate the representativeness of the PWV measurements from the polar orbiting satellites (one per day) taking into account the temporal variability of this meteorological variable.

Technical corrections: There are two typo mistakes in caption of Figures 9 y 10. It says "red bars indicate the classes that are in the 10th percentile" and it should say "blue bars indicate the classes that are in the 10th percentile".

---

## Referee Comment (RC2) · Anonymous Referee #2 · 27 May 2016

This article uses GNSS measurements to assess the diurnal cycle of TCWV and the representativity of satellite measurements that are limited to clear-sky conditions and one overpass per day. I recommend publication after the following minor issues are addressed:

- p.1 l.15: It is not clear that variability mean standard deviation here.

- p.2 l. 12: The influence of precipitation cannot be neglected for MWR measurements.

- p.3 l.3: "can only [be] provided"

- p.4 l.13 / Fig. 2: the text says mean TCWV while the figure caption says median

- p.4 l.15: The meaning of the last sentence is not clear to me. Do you mean: TCWV at 95% of the stations varies between -5% and +5%? - Fig. 8: caption: positive: red, negative: blue

- p.9 l.5: there is a typo: SDS instead of DSD

- p.9 l.17: typo: "where" instead of "were"

- Chapt. 5: Please describe shortly how you decide between clear sky and cloudy cases.

- Fig. 9: "Histogram of [DSD]" instead of bias

- p.11, l.2: "...has to be accounted for [] the interpretation..."

- p.12, l.5: typo: rain gouge instead of rain gauge

---

## Author Comment (AC1) · 22 Jun 2016

Dear Referee#1,

thank you very much for your comments. We corrected the typo.

---

## Author Comment (AC2) · 22 Jun 2016

Dear Referee#2,

thank you very much for you comments. Find below the referee comments in black and the authors response in blue.

- p.1 l.15: It is not clear that variability mean standard deviation here.
We changed that to: "On average, the TCWV standard deviation is about 15% regarding the daily mean."

- p.2 l. 12: The influence of precipitation cannot be neglected for MWR measurements.
We changed the sentence to : "The influence of clouds and precipitation can be neglected for GNSS observations"

- p.3 l.3: "can only [be] provided"
We changed that.

- p.4 l.13 / Fig. 2: the text says mean TCWV while the figure caption says median
We changed that to median in the text.

- p.4 l.15: The meaning of the last sentence is not clear to me. Do you mean: TCWV at 95% of the stations varies between -5% and +5%?
Yes, you are right. Thanks for the suggestion, be adopted it.

- Fig. 8: caption: positive: red, negative: blue
We changed that.

- p.9 l.5: there is a typo: SDS instead of DSD
We changed that.

- p.9 l.17: typo: "where" instead of "were"
Thanks, we changed that.

- Chapt. 5: Please describe shortly how you decide between clear sky and cloudy cases.
We included: "The cloud information was extracted from the operational MERIS cloud mask. For the investigation in this and the next chapter, the corresponding MERIS pixel was spatially collocated for each GNSS observation."

- Fig. 9: "Histogram of [DSD]" instead of bias
We changed that

- p.11, l.2: "...has to be accounted for [] the interpretation..."
We changed it to: "The fact that TCWV derived from instruments like MERIS and MODIS is limited to cloud-free areas has to be considered in the interpretation of trend analysis."

- p.12, l.5: typo: rain gouge instead of rain gauge
We changed that.